# SAFE-DNN: A Deep Neural Network with Spike Assisted Feature Extraction for noise robust inference

## Abstract

We present a Deep Neural Network with Spike Assisted Feature Extraction (SAFE-DNN) to improve robustness of classification under stochastic perturbation of inputs. The proposed network augments a DNN with unsupervised learning of low-level features using spiking neuron network (SNN) with Spike-Time-Dependent-Plasticity (STDP). The complete network learns to ignore local perturbation while performing global feature detection and classification. The experimental results on CIFAR-10 and ImageNet subset demonstrate improved noise robustness for multiple DNN architectures without sacrificing accuracy on clean images.

## 1 Introduction

There is a growing interest in deploying DNNs in autonomous systems interacting with physical world such as autonomous vehicles and robotics. It is important that an autonomous systems make reliable classifications even with noisy data. However, in a deep convolutional neural networks (CNN) trained using stochastic gradient descent (SGD), pixel level perturbation can cause kernels to generate incorrect feature maps. Such errors can propagate through network and degrade the classification accuracy (Nazaré et al. (2017); Luo & Yang (2014)).

Approaches for improving robustness of a DNN to pixel perturbation can be broadly divided into two complementary categories. First, many research efforts have developed image de-noising (or filtering) networks that can pre-process an image before classification, but at the expense of additional latency in the processing pipeline (Ronneberger et al. (2015); Na et al. (2019); Xie et al. (2012); Zhussip & Chun (2018); Soltanayev & Chun (2018); Zhang et al. (2017)). De-noising is an effective approach to improve accuracy under noise but can degrade accuracy for clean images (Na et al. (2019)). Moreover, de-noising networks trained on a certain noise type do not perform well if the a different noise structure is experienced during inference (Zhussip & Chun (2018)). Advanced de-noising networks are capable of generalizing to multiple levels of a type of noise and effective for different noise types (Zhussip & Chun (2018); Soltanayev & Chun (2018); Zhang et al. (2017)). But high complexity of these network makes them less suitable for real-time applications and lightweight platforms with limited computational and memory resources.

An orthogonal approach is to develop a classification network that is inherently robust to input perturbations. Example approaches include training with noisy data, introducing noise to network parameters during training, and using pixel level regularization (Milyaev & Laptev (2017); Nazaré et al. (2017); Luo & Yang (2014); Na et al. (2018); Long et al. (2019)). These approaches do not change the processing pipeline or increase computational and memory demand during inference. However, training-based approaches to design robust DNNs also degrade classification accuracy for clean images, and more importantly, are effective only when noise structure (and magnitude) during training and inference closely match. Therefore, a new class of DNN architecture is necessary for autonomous system that is inherently resilient to input perturbations of different type and magnitude without requiring training on noisy data, as well as computationally efficient.

Towards this end, this paper proposes a new class of DNN architecture that *integrates features extracted via unsupervised neuro-inspired learning and supervised training*. The neuro-inspired learning, in particular, spiking neural network (SNN) with spike-timing-dependent plasticity (STDP) is an alternative and unsupervised approach to learning features in input data (Hebb et al. (1950);

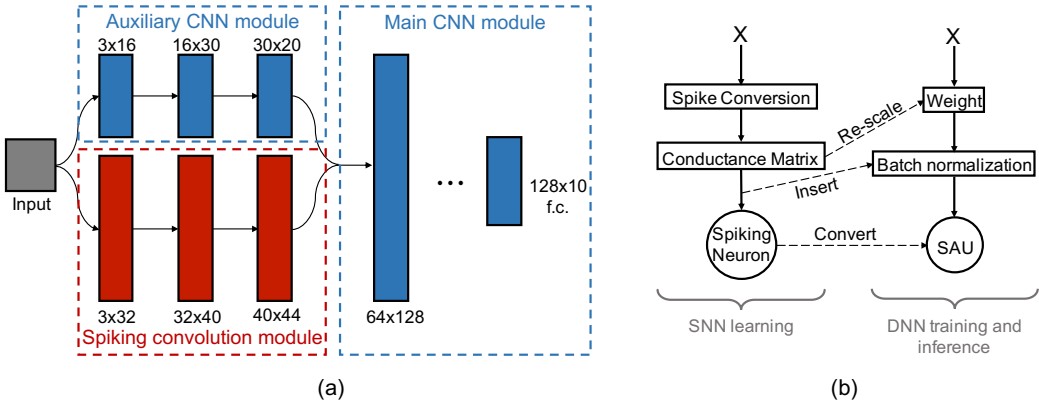

Figure 1: (a) An example architecture of SAFE-DNN. (b) Transition of building blocks from SNN to spiking convolution module of SAFE-DNN, with a special activation unit (SAU)

Bi & Poo (2001); Diehl & Cook (2015); She et al. (2019a); Querlioz et al. (2013); Srinivasan et al. (2016)). STDP based SNN optimizes network parameters according to causality information with no labels (Moreno-Bote & Drugowitsch (2015); Lansdell & Kording (2019)). However, the classification accuracy of a STDP-learned SNN for complex datasets is much lower than a that of a DNN.

The fundamental premise of this paper is that, augmenting the feature space of a supervised (trained) DNN with features extracted by an SNN via STDP-based learning increases robustness of the DNN to input perturbations. We argue that stochastic gradient descent (SGD) based back-propagation in a DNN enables global learning between low-level pixel-to-pixel interactions and high-level detection and classification. On the other hand, STDP performs unsupervised local learning and extracts low-level features under spatial correlation. By integrating features from global (supervised training) and local (STDP) learning, the hybrid network "learns to ignore" locally uncorrelated perturbations (noise) in pixels while extracting the correct feature representation from the overall image. Consequently, hybridization of SGD and STDP enables robust image classification under noisy input while preserving the accuracy of the baseline DNN for clean images.

We present a hybrid network architecture, referred to as Spike Assisted Feature Extraction based Deep Neural Network (SAFE-DNN), to establish the preceding premise. We develop an integrated learning/training methodology to couple the features extracted via neuro-inspired learning and supervised training. In particular, this paper makes the following contributions:

- We present a SAFE-DNN architecture (Figure 1) that couples STDP-based robust learning of local features with SGD based supervised training. This is achieved by integrating a spiking convolutional module within a DNN pipeline.

- We present a novel frequency-dependent stochastic STDP learning rule for the spiking convolutional demonstrating local competitive learning of low level features. The proposed learning method makes the feature extracted by the spiking convolutional module robust to local perturbations in the input image.

- We develop a methodology to transform the STDP-based spiking convolution to an equivalent CNN. This is achieved by using a novel special neuron activation unit (SAU), a non-spiking activation function, that facilitates integration of the SNN extracted features within the DNN thereby creating a single fully-trainable deep network. The supervised (SGD-based) training is performed in that deep network after freezing the STDP-learnt weights in the spiking CNN module.

We present implementations of SAFE-DNN based on different deep networks including MobileNet, ResNet and DenseNet (Sandler et al. (2018), He et al. (2015), Huang et al. (2016)) to show the versatility of our network architecture. Experiment is conducted for CIFRA10 and ImageNet subset considering different types of noise, including Gaussian, Wald, Poisson, Salt&Paper, and adversarial noise demonstrating robust classification under input noise. Unlike training-based approaches, SAFE-DNN shows improved accuracy for a wide range of noise structure and magnitude without requiring any prior knowledge of the perturbation during training and inference and does not degrade the accuracy for clean images (even shows marginal improvement in many cases). SAFE-DNN

complements, and can be integrated with, de-noising networks for input pre-processing. However, unlike de-noising networks, the SAFE-DNN has negligible computation and memory overhead, and does not introduce new stages in the processing pipeline. Hence, SAFE-DNN is an attractive architecture for resource-constrained autonomous platforms with real-time processing.

We note that, SAFE-DNN differs from deep SNNs that convert a pre-trained DNN to SNN (Sengupta et al. (2019), Hu et al. (2018)). Such networks function as a spiking network during inference to reduce energy; however, the learning is still based on supervision and back-propagation. In contrast, SAFE-DNN hybridizes STDP and SGD during learning but creates a single hybrid network operating as a DNN during inference.

## 2 BACKGROUND ON SNN

Spiking neural network uses biologically plausible neuron and synapse models that can exploit temporal relationship between spiking events (Moreno-Bote & Drugowitsch (2015); Lansdell & Kording (2019)). There are different models that are developed to capture the firing pattern of real biological neurons. We choose to use Leaky Integrate Fire (LIF) model in this work described by:

$$dv/dt = a + bv + cI; \text{ and } v = v_{reset}, \text{ if } v > v_{threshold} \tag{1}$$

where, $a$, $b$ and $c$ are parameters that control neuron dynamics, and $I$ is the sum of current signal from all synapses that connects to the neuron.

In SNN, two neurons connected by one synapse are referred to as pre-synaptic neuron and post-synaptic neuron. Conductance of the synapse determines how strongly two neurons are connected and learning is achieved through modulating the conductance following an algorithm named spike-timing-dependent-plasticity (STDP) (Hebb et al. (1950); Bliss & Gardner-Medwin (1973); Gerstner et al. (1993)). With two operations of STDP: long-term potentiation (LTP) and long-term depression (LTD), SNN is able to extract the causality between spikes of two connected neurons from their temporal relationship. More specifically, LTP is triggered when post-synaptic neuron spikes closely after a pre-synaptic neuron spike, indicating a causal relationship between the two events. On the other hand, when a post-synaptic neuron spikes before pre-synaptic spike arrives or without receiving a pre-synaptic spike at all, the synapse goes through LTD. For this model the magnitude of modulation is determined by (Querlioz et al. (2013)):

$$\Delta G_p = \alpha_p e^{-\beta_p (G - G_{min})/(G_{max} - G_{min})} \text{ and } \Delta G_d = \alpha_d e^{-\beta_d (G_{max} - G)/(G_{max} - G_{min})} \tag{2}$$

In the functions above, $\Delta G_p$ is the magnitude of LTP actions, and $\Delta G_d$ is the magnitude of LTD actions. $\alpha_p$, $\alpha_d$, $\beta_p$, $\beta_d$, $G_{max}$ and $G_{min}$ are parameters that are tuned based on specific network configurations.

## 3 MOTIVATION BEHIND SAFE-DNN

The gradient descent based weight update process in a DNN computes the new weight as $W' = W - \eta \nabla L$, where the gradient of loss function $L$ is taken with respect to weight: $\nabla_w L = \langle \frac{\partial L}{\partial W_i}, ..., \frac{\partial L}{\partial W_k} \rangle$. Consider cross entropy loss as an example for $L$, weight optimization of element $i$ is described by:

$$W_i' = W_i - \eta \frac{-\frac{1}{N} \partial \{ \sum_{n=1}^{N} [y_n log(\hat{y}_n)] \}}{\partial W_i} \tag{3}$$

Here $\eta$ is the rate for gradient descent; $N$ is the number of classes; $y_n$ is a binary indicator for the correct label of current observation and $\hat{y}_n$ is the predicated probability of class $n$ by the network. For equation (3), gradient is derived based on the output prediction probabilities $\hat{y}$ and ground truth. Such information is available only at the output layer. To generate the gradient, the output prediction (or error) has to be back-propagated from the output layer to the target layer using chain rule. As $\hat{y} = g(W, X)$ with $g$ being the logistic function and $X$ the input image, the prediction probabilities are the outcome of the entire network structure. Consider the low level feature extraction layers in a deep network. Equation (3) suggests that gradient of the loss with respect to a parameter is affected

by all pixels in the entire input image. In other words, the back-propagation makes weight update sensitive to non-neighboring pixels. This facilitates global learning and improve accuracy of higher level feature detection and classification.

However, the global learning also makes it difficult to strongly impose local constraints during training. Hence, the network *does not learn to ignore* local perturbations during low-level feature extraction as it is trained to consider global impact of each pixel for accurate classifications. This means that during inference, although a noisy pixel is an outlier from the other pixels in the neighbourhood, a DNN must consider that noise as *signal* while extracting low-level features. The resulting perturbation from pixel level noise propagates through the network, and degrades the classification accuracy.

The preceding discussion suggests that, to improve robustness to stochastic input perturbation (noise), the low level feature extractors must learn to consider local spatial correlation. The local learning will allow network to more effectively "ignore" noisy pixels while computing the low-level feature maps and inhibit propagation of input noise into the DNN pipeline.

The motivation behind SAFE-DNN comes from the observation that STDP in SNN enables local learning of features. Compared to conventional DNN, SNN conductance is not updated through gradient descent that depends on back propagation of global loss. Consider a network with one spiking neuron and $n$ connected input synapses, a spiking event of the neuron at time $t_{spike}$ and timing of closest spikes from all input spike trains $T_{input}$, the modulated conductance is given by:

$$G'_i = G_i + \text{sign}(\Delta t_i) \cdot r(G_i) \cdot p(\Delta t_i, f_i) \tag{4}$$

Here $\Delta t_i = t_{spike} - T^i_{input}$ is spike timing difference, $r$ is the magnitude function (Equation 2) and $p$ is the modulation probability function (Equation 5). The value of $t_{spike}$ is a result of the neuron's response to the collective sum of input spike trains in one kernel. Hence, the modulation of weight of each synapse in a SNN depends only on other input signals within the same (local) receptive field. Moreover, as the correlation between the spike patterns of neighboring pre-synaptic neurons controls and causes the post-synaptic spike, STDP helps the network learn the expected spatial correlation between pixels in a local region. During inference, if the input image contains noise, intensity of individual pixel can be contaminated but within a close spatial proximity the correlation is better preserved. As the SNN has learned to respond to local correlation, rather than individual pixels, the neuron's activity experiences less interference from local input perturbation. In other words, the SNN "learns to ignore" local perturbations and hence, the extracted features are robust to noise.

## 4 SAFE-DNN ARCHITECTURE AND LEARNING PROCESS

### 4.1 NETWORK ARCHITECTURE

Fig. 1 (a) shows an illustrative implementation of SAFE-DNN. The network contains spiking layers placed contiguously to form the spiking convolution module, along with conventional CNN layers. The spiking convolution module is placed at the front to enable robust extraction of local and low-level features. Further, to ensure that the low-level feature extraction also considers global learning, which is the hallmark of gradient back-

Table 1: Network Complexity

| Model | Params (M) | MACs (G) |
|---|---|---|
| Baseline MobileNetV2 | 3.50 | 0.33 |
| Baseline ResNet101 | 44.55 | 7.87 |
| Baseline DenseNet121 | 7.98 | 2.90 |
| **SAFE-MobileNetV2** | 3.57 | 0.36 |
| **SAFE-ResNet101** | 44.62 | 7.90 |
| **SAFE-DenseNet121** | 8.04 | 2.94 |

propagation as discussed in section 3, we place several conventional CNN layers of smaller size in parallel with the spiking convolution module. This is called the auxiliary CNN module. The output feature map of the two parallel modules is maintained to have the same height and width, and concatenated along the depth to be used as input tensor to the remaining CNN layers, referred to as the main CNN module. Main CNN module is responsible for higher level feature detection as well as the final classification. The main CNN module can be designed based on existing deep learning models. The concatenation of features from auxilary CNN and spikining convolutional module helps integrate global and local learning.

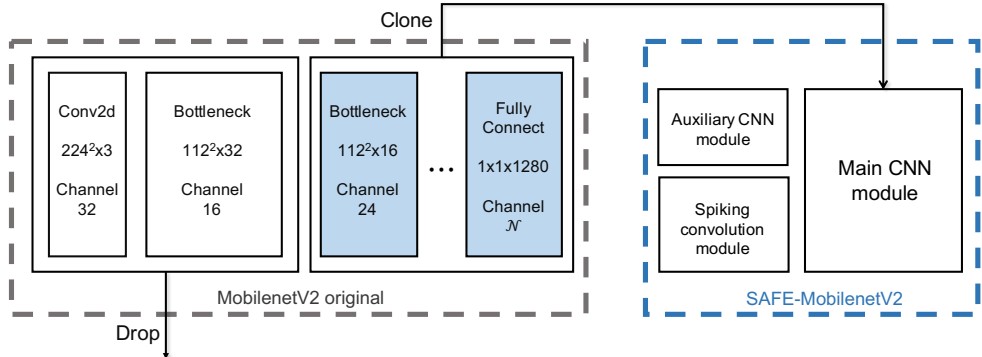

Figure 2: Creating SAFE-MobileNetV2 from the original MobileNetV2

Fig. 2 shows the process of implementing SAFE-MobileNetV2 based on the original MobileNetV2. The first convolution layer and the following one block from the original network architecture are dropped and the remaining layers are used as the mian CNN module of SAFE-MobileNetV2. We show that SAFE-DNN is a versatile network by testing three configurations in this work, which have the main CNN module based on MobileNetV2, ResNet101 and DenseNet121, respectively. The storage and computational complexity of the networks are shown in Table 1. It can be observed that SAFE-DNN implementations do not introduce a significant overhead to the baseline networks.

In the dynamical system of SNN, neurons transmit information in the form of spikes, which are temporally discrete events that spread across multiple simulation time steps. This requires input signal intensity to be converted to spike trains, and a number of time steps for neurons to respond to input stimulus. Such mechanism is different from that of the conventional DNN, which takes only one time step for data to propagate through the network. Due to this reason the native SNN model can not be used in spiking convolution module of SAFE-DNN. Two potential solutions to this problem are, running multiple time steps for every input, or, adapting the spiking convolution module to single-time-step response system. Since the first slows down both training and inference by at least one order of magnitude, we choose the latter.

**Training Process.** We separate STDP-based learning and DNN training into two stages. In the first stage, the spiking convolution module operates in isolation and learns all images in the training set without supervision. The learning algorithm follows our novel frequency dependent STDP method described next in section 4.2. In the second stage, network parameters are first migrated to the spiking convolution module of SAFE-DNN. The network building blocks of the spiking convolutional module go through a conversion process shown in Fig. 1 (b). The input signal to spike train conversion process is dropped, and conductance matrix is re-scaled to be used in the new building block. Batch normalization is inserted after the convolution layer. In order to preserve the non-linear property of spiking neurons, a special activation unit (SAU) is designed to replace the basic spiking neuron model. Details about SAU is discussed later in section 4.3. Once the migration is completed, the entire SAFE-DNN is then trained fully using statistical method, while weights in the spiking convolution module are kept fixed to preserve features learned by SNN. Network inference is performed using the network architecture created during the second stage of training i.e. instead of the baseline LIF, the SAU is used for modeling neurons.

## 4.2 SPIKING CONVOLUTIONAL MODULE

**Frequency-dependent stochastic STDP** The STDP algorithm discussed in 2 captures the basic exponential dependence on timing of synaptic behavior, but does not address the associative potentiation issue in STDP ( Levy & Steward (1979); Carew et al. (1981); Hawkins et al. (1983)). Associativity is a temporal specificity such that when a strong (in case of our SNN model, more frequent) input and a weak (less frequent) input into one neuron induce a post-synaptic spike, a following conductance modulation process is triggered equivalently for the both.

In the context of STDP based SNN, associativity can cause erroneous conductance modulation if unaccounted for (She et al. (2019b)). Therefore, we propose a frequency-dependent (FD) stochastic STDP that dynamically adjust the probability of LTP/LTD based on input signal frequency. The algorithm is described by:

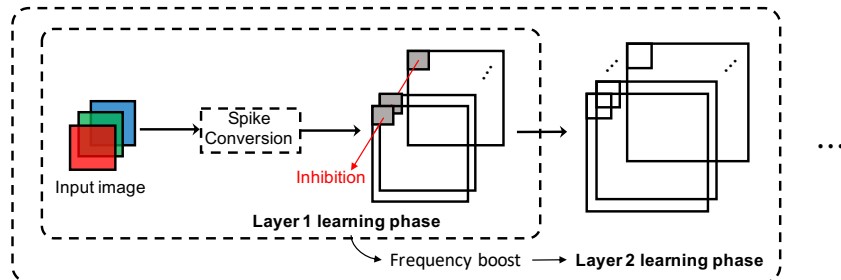

Figure 3: The architecture of the spiking convolutional module for feature extraction and layer-by-layer learning process.

$$P_p = \gamma_p e^{(-\Delta t/(\tau_p(1+\phi_p \frac{f-f_{min}}{f_{max}-f_{min}})))} \text{ and } P_d = \gamma_d e^{(\Delta t/(\tau_d(1+\phi_d \frac{f-f_{min}}{f_{max}-f_{min}})))} \quad (5)$$

In this algorithm, $\tau_d$ and $\tau_p$ are time constant parameters. $\Delta t$ is determined by subtracting the arrival time of the pre-synaptic spike from that of the post-synaptic spike ($t_{post} - t_{pre}$). Probability of LTP $P_p$ is higher with smaller $\Delta t$, which indicates a stronger causal relationship. The probability of LTD $P_d$ is higher when $\Delta t$ is larger. $\gamma_p$ and $\gamma_d$ controls the peak value of probabilities. $f_{max}$ and $f_{min}$ define the upper and lower limit of input spike frequency and $f$ is the value of input spike frequency. When input spike originates from a weak input, the probability declines faster than that from a strong input. As a result, pre-synaptic spike time of weak input needs to be much closer to the post-synaptic spike than that of strong input to have the same probability of inducing LTP, i.e. the window for LTP is narrower for weak input. The same rule applies to LTD behavior. As will be shown in the following section, FD stochastic STDP exhibits better learning capability than conventional STDP.

**SNN architecture** The architecture of the spiking convolutional module is shown in Fig. 3. This architecture resembles conventional DNN but have some differences. First, the 8-bit pixel intensity from input images is converted to spike train with frequency over a range from $f_{min}$ to $f_{max}$. The input spike train matrix connects to spiking neurons in the spiking convolution layer in the same way as conventional 2D convolution, which also applies for connections from one spiking convolution layer to the next. All connections as mentioned are made with plastic synapses following STDP learning rule. When a neuron in the convolution layer spikes, inhibitory signal is sent to neurons at the same (x,y) coordinate across all depth in the same layer. This cross-depth inhibition prevents all neurons at the same location from learning the same feature. Overall, such mechanism achieves a competitive local learning behavior of robust low level features that are crucial to the implementation of SAFE-DNN.

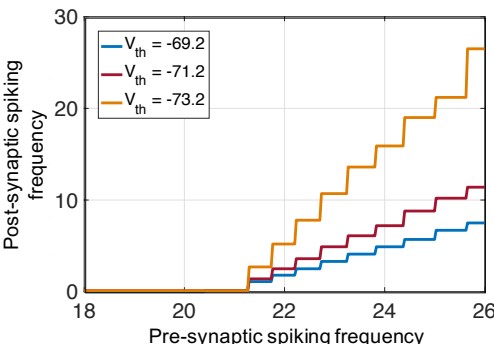

A basic property of spiking neuron is that a number of spikes need to be received before a neuron reaches spiking state and emits one spike. In a two layer network this does not cause a problem but for multiple-layer network it prohibits spiking signal to travel deep down. Due to the diminishing spiking frequency of multiple-layer SNN, a layer-by-layer learning procedure is used. When the first layer completes learning, its conductance matrix is kept fixed and cross-depth inhibition disabled. Next, all neurons in the first layer are adjusted to provide higher spiking frequency by lowering the spiking threshold $V_{th}$. The effect of changing $V_{th}$ is illustrated in Fig.4. In such way, neurons in the first layer receive input from input images and produce enough spikes that can facilitate learning behavior of the second layer. The same process is repeated until all layers complete learning.

Figure 4: Post-synaptic spiking frequency (Hz) vs. pre-synaptic spike frequency (Hz)

### 4.3 SPECIAL ACTIVATION UNIT

Consider the spike conversion process of SNN, given an input value of $X \in [0, 1]$ and input perturbation $\xi$, conversion to spike frequency with range $\epsilon \in [f_{min}, f_{max}]$ is applied such that

$F = Clip_\epsilon\{(X + \xi)(f_{max} - f_{min})\}$. For the duration of input signal $T_{input}$, the total received spikes for the recipient is $N_{spike} = \lfloor F * T_{input} \rfloor$. Also consider how one spiking neuron responses to input frequency variation, which is shown in Fig.4: it can be observed that flat regions exist throughout spiking activity as its unique non-linearity. Therefore, for $|\xi| \leq \frac{\delta}{T_{input}(f_{max} - f_{min})}$ perturbation does not cause receiving neuron to produce extra spikes. While the exact value of $\delta$ changes with different input frequency, it is small only when original input frequency is near the edges of non-linearity. This provides the network with extra robustness to small input perturbations. Based on this, we design the Special Activation Unit (SAU) to be a step function in the form of $f(x) = \sum_{i=1}^{n} \alpha_i \chi_i(x)$ where $\alpha_i$ and $\chi_i$ are pre-defined multiplication parameter and interval indicator function.

## 5 EXPERIMENTAL RESULTS

### 5.1 CIFAR10 DATASET

Three baseline networks: MobileNetV2, ResNet101 and DenseNet121 are tested in comparison with SAFE-DNN. We also studied two enhancement methods for baseline networks, namely, training with noisy input (30 dB) and using average filter (2x2) for image pre-processing. *Note SAFE-DNN is never trained with noisy images; it is only trained with clean images and only tested with noisy images*. Fig. 5 shows training and test loss (top), and training accuracy and test accuracy (bottom) for the training process SAFE-MobileNetV2. The training time using a desktop machine with Intel Core i7-7700K and two NVIDIA GTX 1080 Ti GPUs, SNN learning takes 265 minutes. Training time for SAFE-MobileNetV2 is 63 minutes, for SAFE-ResNet101, 412 minutes and for SAFE-DenseNet121, 274 minutes.

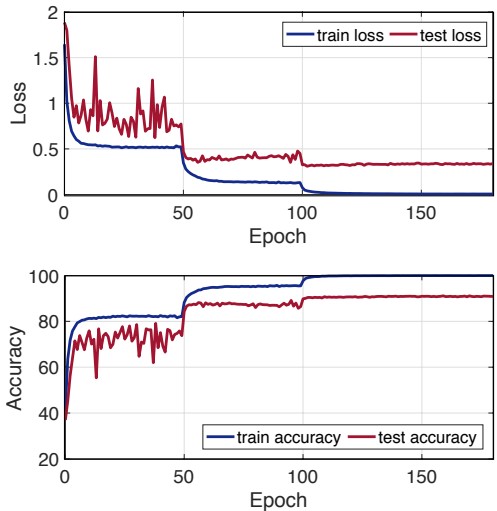

Figure 5: Training accuracy and loss; test accuracy and loss for SAFE-MobileNetV2.

**Visualization of the embedding space** We demonstrate the improved local feature extraction of FD stochastic STDP by comparing capability of the deep network to cluster noisy input. Two SAFE-MobileNetV2 are trained with FD stochastic STDP and deterministic STDP, respectively, and tested on noisy input with AWGN noise. The embedding space is taken between the two fully connected layers and each color represents one class. As shown in Fig. 6, 20 dB input is used for (i) and (ii), 18 dB for (iii) and (iv) and 15 dB for (v) and (vi). SAFE-MobileNetV2 implemented with features extracted via FD stochastic STDP provides better clustering of different classes and achieves higher accuracy.

Next, we compare the entire SAFE-DNN architecture with alternative designs. First, we consider the standard (baseline) MobileNetV2. The second one, referred to as MobileNetV2-$\mu$, has the same architecture as SAFE-MobileNetV2, but the spiking convolution module is replaced with regular trainable DNN layers. The third one, referred to as the MobileNetV2-$\lambda$, is constructed by replacing the activation functions in the first three layers of a trained MobileNetV2-$\mu$ with the SAU (without any re-training). The comparisons with MobileNetV2-$\mu$ and MobileNetV2-$\lambda$ show whether benefits of SAFE-MobilenetV2 can be achieved by only architectural modifications or new (SAU) activation function, respectively, without local STDP learning. All networks are trained with CIFAR10 dataset. Fig. 7 shows embedding space visualizations of all four networks with clean and noisy (SNR equal to 25dB) images. We observe that with clean input images, the vectors in embedding space of the baseline MobileNetV2 are distributed into ten distinct clusters. As noise is added to the images the clusters overlap which leads to reduced classification accuracy. On the other hand, SAFE-MobileNetV2 is able to maintained good separation between feature mappings for each class from no noise to 25 dB. We further observe that clusters for noisy images also heavily

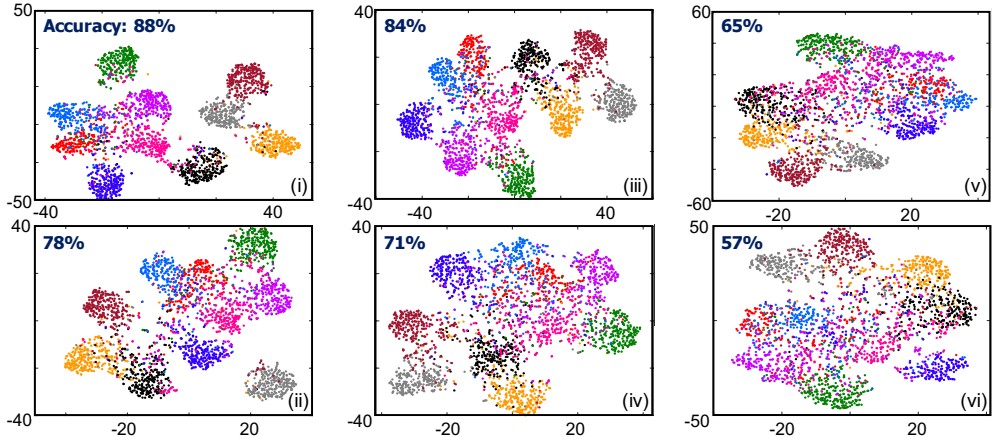

Figure 6: Visualization of the embedding space and classification accuracy from using FD stochastic STDP (top) and deterministic STDP (bottom) learned features

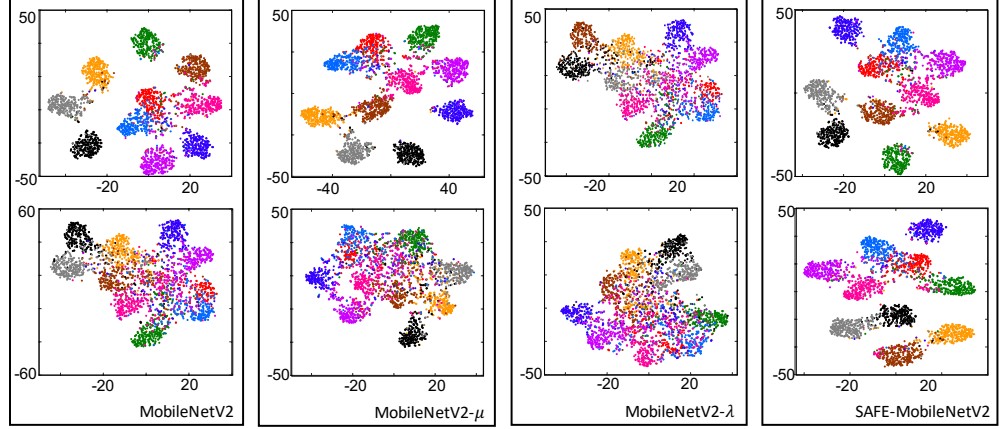

Figure 7: Visualization of the embedding space and classification accuracy from SAFE-MobileNetV2 and three baseline networks for clean (top) and noisy (bottom) input images.

overlap for MobileNetV2-$\mu$ and MobileNetV2-$\lambda$, showing that only using architectural modification or spiking activation function, without STDP learning, cannot improve noise robustness of a DNN.

**Accuracy comparison** Table 2 shows accuracy of all network variants for CIFAR-10. For the baseline DNNs, noise in images significantly degrades classification accuracy. The networks that are trained with noise (30dB noise is used during training) show higher robustness to noise, and the improvement is more prominent when inference noise is at similar level (30 dB) with training noise. For clean images the accuracy is degraded. Average filtering provides accuracy gain over the original networks in highly noisy conditions (less than 20 dB signal to noise ration (SNR)); but major

Table 2: Accuracy (%) results for CIFAR10 with AWGN noise

| Model | Clean | 40 dB | 30 dB | 25 dB | 20 dB | 15 dB | 12 dB |
|---|---|---|---|---|---|---|---|
| Baseline MobileNetV2 | 91.30 | 90.86 | 84.85 | 66.25 | 35.13 | 18.50 | 14.26 |
| Baseline ResNet101 | 93.57 | 89.74 | 86.39 | 78.32 | 55.47 | 26.33 | 15.53 |
| Baseline DenseNet121 | 93.00 | 92.87 | 89.84 | 82.59 | 60.42 | 27.10 | 16.88 |
| Noise trained MobileNetV2 | 90.54 | 90.64 | 90.16 | 86.36 | 62.22 | 25.37 | 16.51 |
| Noise trained ResNet101 | 92.41 | 92.51 | 92.26 | 90.92 | 77.81 | 35.97 | 19.85 |
| Noise trained DenseNet121 | 91.88 | 91.86 | 91.71 | 90.74 | 75.35 | 33.89 | 19.35 |
| MobileNetV2 with average filter | 58.91 | 55.12 | 48.37 | 42.56 | 38.79 | 33.36 | 29.88 |
| ResNet101 with average filter | 60.18 | 57.06 | 49.64 | 45.02 | 39.50 | 34.88 | 32.79 |
| DenseNet121 with average filter | 59.58 | 58.86 | 51.00 | 46.89 | 42.05 | 35.06 | 34.09 |
| **SAFE-MobileNetV2** | 91.33 | 91.25 | 90.01 | 90.68 | 87.88 | 64.95 | 39.22 |
| **SAFE-ResNet101** | 93.59 | 93.43 | 92.13 | 92.11 | 90.47 | 70.85 | 43.25 |
| **SAFE-DenseNet121** | 93.03 | 92.86 | 92.70 | 91.35 | 88.00 | 62.99 | 33.19 |

Table 3: Top 1 Accuracy (%) results for ImageNet subset with noise

| Model | Clean | 25 dB | 15 dB | 10 dB | 5 dB |
|---|---|---|---|---|---|
| Baseline MobileNetV2 | 70.80 | 67.41 | 57.92 | 45.35 | 34.48 |
| Baseline ResNet101 | 71.02 | 67.81 | 64.04 | 46.27 | 35.47 |
| Baseline DenseNet121 | 70.92 | 67.60 | 63.28 | 44.34 | 27.50 |
| Noise trained MobileNetV2 | 66.30 | 68.12 | 59.71 | 46.70 | 34.66 |
| Noise trained ResNet101 | 68.91 | 69.20 | 65.71 | 52.60 | 41.32 |
| Noise trained DenseNet121 | 69.13 | 70.51 | 66.47 | 52.76 | 36.32 |
| MobileNetV2 with average filter | 67.44 | 66.91 | 61.69 | 51.69 | 40.43 |
| ResNet101 with average filter | 68.18 | 68.25 | 65.14 | 53.09 | 41.50 |
| DenseNet121 with average filter | 65.38 | 64.56 | 62.40 | 50.65 | 39.40 |
| **SAFE-MobileNetV2** | 71.05 | 67.86 | 65.91 | 53.82 | 42.33 |
| **SAFE-ResNet101** | 71.14 | 70.67 | 67.24 | 55.04 | 42.87 |
| **SAFE-DenseNet121** | 70.81 | 69.44 | 65.47 | 54.30 | 40.84 |

performance drop is observed under mild to no noise. This is expected as average filtering results in significant loss of feature details for input images in the CIFAR-10 dataset.

For SAFE-DNN implemented with all three DNN architectures, performance in noisy condition is improved over the original network by an appreciable margin. For example, at 20 dB SNR SAFE-MobileNetV2 remains at good performance while the original network drops below 40% accuracy, making a significant (50%) gain. Similar trend can be observed for other noise levels. Compared to networks trained with noise, SAFE-DNN shows similar performance at around 30 dB SNR while its advantage increases at higher noise levels. Moreover, for clean images accuracy of SAFE-DNN is on par with the baseline networks.

## 5.2 TEST ON IMAGENET SUBSET

Considering the use case scenario of autonomous vehicles, we conduct test on a subset of ImageNet that contains classes related to traffic (cars, bikes, traffic signs, etc).The subset contains 20 classes with a total of 26,000 training images. The same baseline networks as in the CIFAR10 test are used. Here 25 dB SNR images are used for noise

Table 4: Top 5 Accuracy (%) results for MobileNetV2 on ImageNet subset with Noise

| Model | Clean | 10 dB | 5 dB |
|---|---|---|---|
| Baseline | 94.72 | 79.20 | 67.15 |
| Noise trained | 92.43 | 81.63 | 68.36 |
| Average filtering | 92.81 | 85.37 | 78.95 |
| **SAFE-MobileNetV2** | 95.91 | 89.57 | 83.92 |

training. The accuracy result is shown in Table 3. All networks achieve around 70% top 1 accuracy on clean images. Noise training shows robustness improvement over the baseline network but still negatively affects clean image accuracy. In this test the average filter shows less degradation under no noise condition than for the CIFAR10 test, due to higher resolution of input images. DensNet121 shows more noise robustness than MobileNetV2 and ResNet101 when noise training is used, while for average filtering ResNet101 benefits the most. SAFE-DNN implementations of all three networks exhibit same or better robustness over all noise levels. Clean image classification accuracy is also unaffected. Comparing top 5 accuracy result for SAFE-MobileNetV2 and its baselines, as shown in Table 4, SAFE-MobileNetV2 is able to maintain above 80% accuracy even at 5 dB SNR, outperforming all three baselines.

## 5.3 MORE PERTURBATION STRUCTURES

**Random perturbation** We test SAFE-DNN in three more noise structures: Wald, Poisson and salt-and-pepper (SP). For CIFAR10, the result is shown in 5. Wald I has a distribution of $\mu = 3, scale = 0.3$ and for Wald II, $\mu = 13, scale = 1$; Poisson I has a distribution with peak of 255 and for Poisson II, 75; S&P I has 5% noisy pixels and S&P II has 20%. Noise-trained DNNs for Wald, Poisson, and SP, are trained with noisy images generated using distributions Wald I, Poisson I, and SP I, resepectively. It can be observed that SAFE-DNN implementation with all three networks are more noise robust than baseline and average filtering. The noise-trained networks performs well when inference noise is aligned with training noise, but performance drops when noise levels are not aligned. Moreover, noise-trained networks trained with mis-aligned noise types performs poorly

Table 5: Accuracy (%) results for CIFAR10 with different noise types

| Model | Wald I | Wald II | Poisson I | Poisson II | S&P I | S&P II |
|---|---|---|---|---|---|---|
| Baseline MobileNetV2 | 83.81 | 48.41 | 63.88 | 37.70 | 68.86 | 32.37 |
| Baseline ResNet101 | 85.12 | 60.75 | 76.49 | 56.47 | 79.25 | 47.69 |
| Baseline DenseNet121 | 87.34 | 63.24 | 78.15 | 60.60 | 83.50 | 49.54 |
| Noise trained MobileNetV2 | 90.06 | 71.80 | 87.18 | 74.51 | 88.86 | 63.84 |
| Noise trained ResNet101 | 91.95 | 88.15 | 89.63 | 81.16 | 90.78 | 74.43 |
| Noise trained DenseNet121 | 91.48 | 84.42 | 89.44 | 78.90 | 89.93 | 72.24 |
| MobileNetV2 w/ average filter | 53.12 | 36.70 | 54.40 | 35.84 | 51.35 | 31.29 |
| ResNet101 w/ average filter | 54.06 | 38.03 | 56.75 | 38.59 | 52.19 | 34.25 |
| DenseNet121 w/ average filter | 55.86 | 39.69 | 56.31 | 37.34 | 51.61 | 32.47 |
| **SAFE-MobileNetV2** | 90.46 | 88.50 | 84.35 | 82.58 | 89.17 | 80.83 |
| **SAFE-ResNet101** | 92.97 | 90.75 | 88.91 | 87.22 | 90.53 | 86.75 |
| **SAFE-DenseNet121** | 92.83 | 89.52 | 87.67 | 85.75 | 90.28 | 85.38 |

Table 6: Accuracy (%) results for ImageNet subset with different noise types

| Model | Wald I | Wald II | Poisson I | Poisson II | S&P I | S&P II |
|---|---|---|---|---|---|---|
| Baseline MobileNetV2 | 68.56 | 58.99 | 65.16 | 51.08 | 66.53 | 46.81 |
| Noise trained MobileNetV2 | 69.07 | 62.12 | 67.31 | 60.22 | 67.99 | 53.86 |
| MobileNetV2 w/ average filter | 66.05 | 61.46 | 64.52 | 59.91 | 62.07 | 45.50 |
| **SAFE-MobileNetV2** | 70.19 | 64.37 | 66.74 | 63.15 | 67.10 | 58.91 |

(results not shown). As for ImageNet subset, networks based on MobileNetV2 are tested. Wald I is a distribution with $\mu = 5, scale = 0.3$ and Wald II is a distribution with $\mu = 25, scale = 1$; Poisson I has a distribution with peak of 255 and for Poisson II, 45; S&P I has 5% noisy pixels and S&P II has 30%. As previously, noise-trained networks are trained with noisy images generated from Wald I, Poisson I and SP I. Similar to previous results, as shown in 6 SAFE-MobileNetV2 is more robust to the different noise structures without ever-being trained on any noise structure.

**Adversarial perturbation** We also test SAFE-DNN on adversarial perturbation crafted from black-box adversarial method. For this test, DNNs trained with conventional method are used as target network to generate the perturbed images. The attack method is fast gradient sign method (Goodfellow et al. (2014)): $X^{adv} = X + \epsilon \; \text{sign}(\nabla_X J(X, y_{true}))$. Here $X$ is the input image, $y_{true}$ the ground truth label and $\epsilon \in [0, 255]$. For source networks that are tested on the perturbed images, DNN trained

Table 7: Accuracy (%) results for CIFAR10 with adversarial perturbation

| Model | $\epsilon = 3$ | $\epsilon = 8$ | $\epsilon = 16$ |
|---|---|---|---|
| Baseline MobileNetV2 | 83.50 | 67.59 | 47.93 |
| Baseline ResNet101 | 85.31 | 68.42 | 47.25 |
| Baseline DenseNet121 | 88.00 | 76.18 | 64.63 |
| **SAFE-MobileNetV2** | 87.14 | 74.26 | 59.18 |
| **SAFE-ResNet101** | 89.14 | 74.67 | 61.24 |
| **SAFE-DenseNet121** | 90.13 | 82.86 | 76.54 |

with different initialization are used as baseline against SAFE-DNN implementation of the deep network. As shown in Table 7, SAFE-DNN also shows improved robustness to noise generated via adversarial perturbations. However, we note that the results do not indicate robustness to white-box attacks; and integration of SAFE-DNN with adversarial training approaches will be an interesting future work in this direction.

## 6 CONCLUSIONS

In this paper we present SAFE-DNN as a deep learning architecture that integrates spiking convolutional network with STDP based learning into a conventional DNN for robust low level feature extraction. The experimental results show that SAFE-DNN improves robustness to different input perturbations without any prior knowledge of the noise during training/inference. SAFE-DNN is compatible with various DNN designs and incurs negligible computation/memory overhead. Hence, it is an attractive candidate for real-time autonomous systems operating in noisy environment.

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
