# OpenReview forum: "SAFE-DNN: A Deep Neural Network with Spike Assisted Feature Extraction for Noise Robust Inference"
_ICLR.cc/2020/Conference — Reject_

### Official Review · AnonReviewer2 · 2019-10-25
**Official Blind Review #2**

**Rating:** 3

**Review:**

The paper develops a method to augment deep neural networks with Spike-Time-Dependent-Plasticity (STDP) aiming at improving noise robustness of the network learned features. In the hybrid network, learned feature is the concatenation of traditionally supervised-ly learned features and those from an auxiliary module trained locally and unsupervised-ly by STDP. The new network demonstrates improved noise robustness via improved classification accuracy on Cifar10 and ImageNet subset when input data have noise, on different network architectures.

The paper, however, fails to address the many works in the literature about adversarial perturbations ('attack') and adversarial training ('defense'), starting by (Szegedy et al., 2013). The different types of attacks affect the efficiency of defense due to the game-theoretical nature of the adversarial perturbation problem. If the attack is blind to the classification model, e.g., Gaussian attack by adding Gaussian noise, then image restoration techniques like denoising could provide an effective 'defense'. Thus model-specific attacks are of more application interest than model-blind ones. The current manuscript did not address the specific noise type being used to perturb the image. It is unlikely that the local learning techniques proposed in the paper can work on many kinds of perturbations especially the 'attacks' which is model specific.

The proposed methodology is a feature concatenation of local (low-level) features of image data and deep features. Given the current state of the manuscript, the level of methodological novelty and the scope of input perturbations that can be made robust against both appear to be limited.

References:
Christian Szegedy, Wojciech Zaremba, Ilya Sutskever, Joan Bruna, Dumitru Erhan, Ian Goodfellow, and Rob Fergus. Intriguing properties of neural networks. arXiv preprint arXiv:1312.6199, 2013.


**Experience Assessment:**

I have read many papers in this area.

**Review Assessment: Checking Correctness Of Derivations And Theory:**

I assessed the sensibility of the derivations and theory.

**Review Assessment: Checking Correctness Of Experiments:**

I assessed the sensibility of the experiments.

**Review Assessment: Thoroughness In Paper Reading:**

I read the paper at least twice and used my best judgement in assessing the paper.

---

> ### Author Response · Authors · 2019-11-15
> **Reply to Review #2**
>
> Thank you for your review.
>
> >>Key contribution of the paper
>
> As mentioned in response to the reviewer #3, the key novelty of the paper is the hybrid architecture that couple features learned via STDP and trained via SGD to create a single classification network. During inference, the hybrid network acts as a DNN and shows better classification accuracy for clean images, as well as improved robustness against input perturbation (noise). To the best of our knowledge, this is first ever demonstration of a DNN, where hybridization of STDP and SGD is performed during training.
>
> >>Comparison to De-noising
>
> Please see response to reviewer #2 on the qualitative comparison with de-noising network as well as noise-trained DNNs.
>
> >>Scope of the Input Noise Type
>
> We conducted more experiments on other noise distributions including: Wald, Poisson, and Salt-and-pepper. Moreover, based on the reviewer’s suggestion we have also included adversarial noise generated via black-box attack. The results are included in the revised paper, and showed that SAFE-DNN, trained only on clean images, show appreciable robustness to different types of noise structures and magnitude.
>
> >>Relation to adversarial noise and “attacks”
>
> The paper is primarily focused on noise or perturbation in the input image that can occur naturally, for example, due to sensor hardware or capturing environment. We acknowledge that there also exists targeted perturbations such as adversarial attacks. We have included new test results considering adversarial noise generated via black-box attack methods and demonstrated that SAFE-DNN can improve robustness to adversarial noise as well.
>
> We would like to stress that the paper does not claim to address the domain of adversarial attack. In particular, current form of SAFE-DNN is vulnerable to white-box attack, as it is possible to extract gradient information from the SAFE-DNN architecture. That being said, the well-studied defense methods such as adversarial training can, in principle, be adapted for STDP learning and deep network training stage, which can improve robustness against white-box attack. However, designing a SAFE-DNN for adversarial attack requires complementary efforts including extensive studies and experimental results, and is outside the scope of the current paper.

---

### Official Review · AnonReviewer1 · 2019-10-27
**Official Blind Review #1**

**Rating:** 6

**Review:**

The paper shows that replacing feature extraction layers by spiking convolution network can improve the performance under random noise. The algorithm itself is simple since it's just a combiniation of STDP and standard CNN. The results shows improved performance under some random noise. Although the idea is cute, I feel the paper fails to convince why spiking nets are more robust to random noise; the explanation using backprop rules in section 3 sounds interesting but does not fully convince me; for example, if we train a CNN by other approach instead of back-propagation, can we also improve robustness to input noise? Also, what kind of input noise are we considering in the analysis?

Also, I have some questions on the experiments:

1. Experiments are only tested under one kind of random perturbation with different strengths. I think it will be better if the algorithm can consistently improve over various kinds of noise distributions.

2. It is mentioned in the introduction that some methods were proposed to filter out the input noise, but they are not compared in the experiments.

3. What's the training time of the proposed method?

**Experience Assessment:**

I have published in this field for several years.

**Review Assessment: Checking Correctness Of Derivations And Theory:**

I assessed the sensibility of the derivations and theory.

**Review Assessment: Checking Correctness Of Experiments:**

I assessed the sensibility of the experiments.

**Review Assessment: Thoroughness In Paper Reading:**

I read the paper at least twice and used my best judgement in assessing the paper.

---

> ### Author Response · Authors · 2019-11-15
> **Reply to Review #1**
>
> Thank you for your review.
>
> >> Novelty in Combining STDP and CNN.
>
> We agree (and like) that the concept is simple where STDP-learned and SGD-trained features are combined to provide improved classification performance under input perturbation. However, as mentioned in the previous response, this was achieved through innovation in (i) STDP algorithm to improved robustness against input perturbation; and (ii) hybridization to seamlessly integrate STDP-learned features into the DNN pipeline during training and inference. It is important to realize that the hybrid network behaves as a DNN during inference and there are no spiking neurons during inference.
>
> >> Explanation of the noise robustness of SNN learned features:
>
> We have improved section 3 and hope it provides a better reasoning for the robustness of low level features extracted by SNN.
>
> We are intrigued by the reviewer’s question on whether a normal CNN trained using methods other than back propagation will show similar noise robustness or not. In principle, we feel an alternative to back-propagation that does not distribute errors globally during training may achieve a similar objective; however, at this point we do not have any empirical evidence and quantitative result to prove (or dis-prove) this hypothesis.
>
> >> What kind of input noise are we considering in the analysis?
>
> The initial paper considered pixel level noise modeled as a Gaussian variable.
>
> >> Experiments are only tested under one kind of random perturbation with different strengths. I think it will be better if the algorithm can consistently improve over various kinds of noise distributions.
>
> We conducted more experiments on other noise distributions including: Wald, Poisson and salt-and-pepper. Moreover, based on the suggestion from Reviewer #2, we have also included adversarial noise generated via black-box attack. The results are included in the revised paper, and showed that SAFE-DNN, trained only on clean images, show appreciable robustness to different types of noise structures and  magnitude.
>
> >> It is mentioned in the introduction that some methods were proposed to filter out the input noise, but they are not compared in the experiments.
>
> We have considered one popular approach for noise removal (model-based), namely, average filtering. As observed, the filtering helps for noisy images but significantly degrade the quality for clean images. Please see the response to reviewer #3 on the qualitative discussions on SAFE-DNN versus input denoising.
>
> >>What's the training time of the proposed method?
>
> We have included training time for both SNN and SAFE-DNN implementations in section 4: “… for CIFAR10, using a desktop machine with Intel Core i7-7700K and two NVIDIA GTX 1080 Ti GPUs, SNN simulation takes 265 minutes. Training time for SAFE-MobileNetV2 is 63 minutes, for SAFE-ResNet101, 412 minutes and for SAFE-DenseNet121, 274 minutes. ”

---

### Official Review · AnonReviewer3 · 2019-11-02
**Official Blind Review #3**

**Rating:** 3

**Review:**

The paper proposes a hybrid network architecture that can integrate features extracted via supervised training and unsupervised neuro-inspired learning. The paper is well-written and the experimental results seem sensible. The experimental results mainly revolve around testing the networks over noise added to training images.  The problem of image denoising is very well-studied and very good methods have been proposed for image denoising under arbitrary noise using deep learning (see the works in CVPR, ICCV, ECCV etc.). Unfortunately, I am not in the position to judge the novelty
wrt spiking neuron network literature. Nevertheless, as far as computer vision or general applications is concerned the proposed pipeline would not be among the methods of choice.  Hence, I am recommending weak reject for now, waiting for a more informed opinion to see if I will change my opinion.

**Experience Assessment:**

I do not know much about this area.

**Review Assessment: Checking Correctness Of Derivations And Theory:**

I assessed the sensibility of the derivations and theory.

**Review Assessment: Checking Correctness Of Experiments:**

I assessed the sensibility of the experiments.

**Review Assessment: Thoroughness In Paper Reading:**

I read the paper at least twice and used my best judgement in assessing the paper.

---

> ### Author Response · Authors · 2019-11-15
> **Reply to Review #3**
>
> Thank you for your review.
>
> >>Innovation with respect to SNN Literature
>
> Spiking neural network (SNN) is an attracting idea of realizing biologically plausible neural networks and have been widely studied. However, SNN based on pure STDP learning have yet to show comparable performance as DNN, in particular for complex datasets like ImageNet. More recently, there have been many efforts in realizing supervised training in SNN via back propagation; which can achieve comparable performance as DNN. These prior works on SNN-DNN conversion focus on generating a deep network with spiking activation function (primarily to save energy) but does not have unsupervised learning.
>
> To the best of our knowledge, the proposed method is the first effort in creating a hybrid network that successfully couples supervised training in DNN with local unsupervised learning in SNN in a single architecture. Note, the hybrid network presented here is not a simple cascade of SNN and DNN where SNN acts as a pre-processor. Instead, we propose a tighter coupling of the two by integrating features learned via STDP and features trained using SGD into a single model. The unsupervised learning presented here goes beyond traditional STDP. We present an innovative frequency dependent stochastic STDP formulation that improves ability of the network to extract features that are robust to local perturbation. The additional results are added to illustrate this advantage. The local and cross-depth inhibition, a relatively new concept in the STDP-based learning, has also been incorporated.
>
> After the STDP-learning is completed for the SNN component, we present a new conversion approach, where the SNN architecture is converted to an equivalent DNN, by (i) removing input spike generation, (ii) re-scaling of weights, and (iii) converting the spiking activation function to a special activation function. The conversion allows the hybrid network to be trained as a DNN preserving the accuracy for baseline images.
>
> In summary, in contrast to prior works on ANN to SNN conversion that essentially create an SGD-trained deep SNN with spiking activation; this paper creates a final hybrid network that behaves as a regular DNN during inference but hybridizes STDP and SGD during training to enhance learning capability of the network.
>
> >>Method of choice for noise-robust computer vision
>
> The noise-robust classification can be achieved using two complementary approaches: (i) pre-processing the input via de-noising networks, and (ii) improving robustness of the classification network against input perturbation. SAFE-DNN is an approach for (ii) and can be integrated with techniques for (i) as SAFE_DNN does not degrade accuracy for clean images.
>
> We acknowledge that using deep learning techniques for image de-noising is a well-studied area, and many of them show very good performance for arbitrary noise structures. However, de-noising adds a new stage in the processing pipeline increasing the overall latency. Moreover, for light-weight de-noising networks, if there is a significant difference between noise structure during training and inference the quality degrades (Na et. al., Xie et. al). Also, as de-noising changes the input structure, it can degrade accuracy for clean images. Advanced networks have been proposed to generalize well for different noise levels but their complexity is high increasing complexity of the overall system. In contrast, SAFE-DNN introduces negligible overhead. For example, even a lightweight de-noising network shown by Na, et. al., the overhead is 2.5 times more parameters (0.187M versus 0.07M).
>
> The direct comparison of SAFE-DNN will be with other approaches for (ii), for example, networks trained with noisy images as presented in the paper. In that sense, SAFE-DNN learns to become robust without ever being trained on the noisy images which ensures the network is easily generalized to very different noise structures. We have added additional results in the paper to clearly demonstrate the ability of the network to show robustness to noise of different magnitude and structures. This is the most important advantage of SAFE-DNN over noise-trained DNNs.
>
> We note that, the proposed network is not in conflict with pre-processing techniques such as de-noising DNNs, meaning that SAFE-DNN can be considered as an addition to de-noising. We are currently implementing advanced de-noising networks into our pipeline to show how an integrated pre-processing + SAFE-DNN will perform, and new results will be added to the final paper, if accepted.
>
> Taesik Na, et. al. Noise-robust and resolution-invariant imageclassification with pixel-level regularization.  InInternational Conference on Acoustics, Speechand Signal Processing,(ICASSP), 2018
>
> Junyuan   Xie, et. al. Image   denoising   and   inpainting   withdeep  neural  networks. Advances   in   Neural   Information   Processing   Systems   25,pp.   341–349.   Curran   Associates,    Inc.,  2012

---

### Public Comment · ~Daiheng_Gao2 · 2019-09-29
**A spell error.**

Hi, when I read your paper, I occasionally found that on page 4. there is STPD instead of STDP,  so it may be a little mistake or something?

---

> ### Author Response · Authors · 2019-09-29
> **Re: A spell error.**
>
> Yes it should be STDP. Thank you for pointing that out. I apologize for the confusion.

---

### Author Response · Authors · 2019-11-15
**Revision of script**

1.	We have modified the introduction to better articulate the contribution of the work in the context of prior works in noise-robust DNNs,  as well as, prior works in SNN.
2.	We have added additional details on prior works on noise-robustness including de-noising, and explained the difference between proposed approach and image de-noising.
3.	The text in section 3 is modified to better explain the impact of STDP on improving robustness to noise.
4.	We have modified Section 4 to explain the contribution/novelty of the work compared to prior works in SNN.
5.	Based on the reviewers’ suggestions, more experimental results are included in section 5. Those results are:
    a.	Embedding space visualization comparison for FD stochastic STDP and deterministic STDP to illustrate the role of novel STDP learning techniques proposed in this work.
    b.	Training time analysis of SAFE-DNN.
    c.	Test on additional noise types and structures.
    d.	Test on adversarial noise generated using black-box attacks.

---

### Decision · Program_Chairs · 2019-12-19

**Decision:**

Reject

**Comment:**

 The paper proposes to improve noise robustness of the network learned features, by augmenting deep networks with Spike-Time-Dependent-Plasticity (STDP). The new network show improved noise robustness with better classification accuracy on Cifar10 and ImageNet subset when input data have noise. While this paper is well written, a number of concerns are raised by the reviewers. They include that the proposed method would not be favored from computer vision perspective, it is not convincing why spiking nets are more robust to random noises, and the method fails to address works in adversarial perturbations and adversarial training. Also, Reviewer #2 pointed out the low level of methodological novelty. The authors provided response to the questions, but did not change the rating of the reviewers. Given the various concerns raised, the ACs recommend reject.